# Evaluation of Structured, Semi-Structured, and Free-Text Electronic Health Record Data to Classify Hepatitis C Virus (HCV) Infection

Allan Fong [1] , Justin Hughes [2], Sravya Gundapenini [1,3], Benjamin Hack [4], Mahdi Barkhordar [2], Sean Shenghsiu Huang [5], Adam Visconti [2,6,*], Stephen Fernandez [1] and Dawn Fishbein [1,7]

1   MedStar Health Research Institute, Hyattsville, MD 20782, USA; allan.fong@medstar.net (A.F.)
2   MedStar Health, Columbia, MD 20037, USA
3   School of Medicine, Ross University, Miramar, FL 33027, USA
4   School of Medicine, Georgetown University, Washington, DC 20007, USA
5   Department of Health Management and Policy, School of Health, Georgetown University, Washington, DC 20007, USA
6   Department of Family Medicine, MedStar Georgetown University, Washington, DC 20010, USA
7   MedStar Washington Hospital Center, Washington, DC 20010, USA
*   Correspondence: adam.x.visconti@gunet.georgetown.edu

**Abstract:** Evaluation of the United States Centers for Disease Control and Prevention (CDC)-defined HCV-related risk factors are not consistently performed as part of routine care, rendering risk-based testing susceptible to clinician bias and missed diagnoses. This work uses natural language processing (NLP) and machine learning to identify patients who are at high risk for HCV infection. Models were developed and validated to predict patients with newly identified HCV infection (detectable RNA or reported HCV diagnosis). We evaluated models with three types of variables: structured (structured-based model), semi-structured and free-text notes (text-based model), and all variables (full-set model). We applied each model to three stratifications of data: patients with no history of HCV prior to 2020, patients with a history of HCV prior to 2020, and all patients. We used XGBoost and ten-fold C-statistic cross-validation to evaluate the generalizability of the models. There were 3564 unique patients, 487 with HCV infection. The average C-statistics on the structured-based, text-based, and full-set models for all the patients were 0.777 (95% CI: 0.744–0.810), 0.677 (95% CI: 0.631–0.723), and 0.774 (95% CI: 0.735–0.813), respectively. The full-set model performed slightly better than the structured-based model and similar to text-based models for patients with no history of HCV prior to 2020; average C-statistics of 0.780, 0.774, and 0.759, respectively. NLP was able to identify six more risk factors inconsistently coded in structured elements: incarceration, needlestick, substance use or abuse, sexually transmitted infections, piercings, and tattoos. The availability of model options (structured-based or text-based models) with a similar performance can provide deployment flexibility in situations where data is limited.

**Keywords:** machine learning; XGBoost; natural language processing; text mining; hepatitis C virus; HCV; screening and prevention





## 1. Introduction

Identifying targeted, innovative, and effective methods to screen and treat people living with HCV will contribute towards diagnosing the millions who are not yet identified, and realizing the World Health Organizational goal of eliminating HCV by 2030 [1]. The disparate resources and limited regional and community infrastructure coupled with an often marginalized and socio-economically unstable patient population results in the need to develop multi-faceted and tailored approaches to elimination.

The US rate of new HCV cases has tripled from 2005–2015, ref. [2] primarily due to the rise in opioid use, with resulting calls to broaden HCV testing and treatment [3]. The World

Health Organization has called for the global elimination of HCV (WHO elimination) [1], though to meet this goal, persons at risk of infection need to get tested and, if diagnosed, treated. Traditional risk-based screening alone has failed to capture testing in important populations making testing less effective [4,5]. The American Association for the Study of Liver Diseases and Infectious Diseases Society of America (AASLD/IDSA), the United States Preventive Services Task Force (USPSTF) and the Centers for Disease Control and Prevention (CDC) expanded their recommendations in 2020 to include the HCV screening of all pregnant women and the one-time universal HCV screening in all persons over 18 and repeat testing in those at-risk (current or historic injection drug users, intranasal drug use; HIV diagnosis; long-term hemodialysis; received clotting factor produced prior to 1987; abnormal alanine transaminase (ALT); solid organ donor; prior recipient of a transfusion or organ transplant; exposure to needle sticks, sharps; mucosal exposure to HCV+ blood; ever incarcerated; child born to HCV+ mother; and sexually active persons starting pre-exposure prophylaxis) [6,7]. Previously the recommendation included persons born between 1945 and 1965 and risk-based testing.

There is increasing research in mining electronic health record (EHR) data to support the identification and treatment of HCV patients. Some have demonstrated the utility of using rule-based logic to query EHR data to identify diagnosed but untreated HCV-RNA detectable patients. The output of the algorithms was used by patient navigators to contact potential candidates for care coordination programs [8,9]. While others evaluated various machine learning (ML) algorithms and techniques to predict HCV from an Egyptian patient database using structured elements, i.e., those that are stored in the EHR as discrete elements that map a unique set of variables [10].

Structured elements are limited to what gets coded; however, semi-structured and free-text elements have the advantages of potentially capturing more subtle concepts in the provider notes and narratives. Natural language processing (NLP) is the research and application of computational techniques to extract, model, and analyze written or spoken language [11,12]. NLP and text mining techniques leverage the power of computers to process and make sense of large amounts of text that may otherwise not be input as structured data. Researchers have applied NLP to the analysis of clinical documentation, such as discharge summaries and radiology interpretations, to improve care and workflow processes [13]. NLP can extract concepts as variables that are then used in the development of machine learning models [14,15]. With the increased use of electronic health records, there has been a growing corpus of text information in healthcare that is ripe for NLP and ML.

Despite efforts to improve targeted risk-based testing, evaluation of the CDC-defined HCV-related risk factors are not consistently performed as part of routine care, rendering risk-based testing susceptible to clinician bias and missed diagnoses. Although one-time universal testing has been recommended since 2020 to capture missed diagnoses, [3] persons who are at high-risk for HCV infection need to have repeat testing with continued risk, as well as re-testing after treatment, especially if we are to reach HCV *elimination* by 2030 [1]. One of these approaches should be micro-elimination strategies at larger health-care systems, which can then be widely disseminated to other organizations.

This work aims to develop a ML model to identify patients at high risk for newly identified HCV infections and those who are not easily identified and repeatedly tested by providers. In this study, we use both ML and NLP to predict patients with HCV infections with EHR data. The contributions of this study are two-fold. First, we evaluate and compare the utility of structured, semi-structured, and free-text EHR data to predict patients with HCV infections. Second, we compare prediction models for patients with and without a prior history of HCV.

## 2. Methods

*Data Source and Study Design*

Our study uses data between January 2020 to October 2021 from a ten-hospital academic health system in the mid-Atlantic region of the United States. Models were developed and validated to predict patients with HCV infection (defined as having a detectable RNA or reported HCV diagnosis) amongst patients who had either an antibody (ATB) or RNA test during the study dates. Variables extracted from patient records include year of birth, sex (female or male), and race (Black, White, or Other), ethnicity (Hispanic or Not Hispanic), as well as clinical notes, diagnosis, and medications (specifically, prescribed opioids). Patient records were excluded if they were missing sex, race, ethnicity, ICD10 or notes data. Patients were further stratified if they had a history of HCV prior to 2020 as defined by ICD10 codes (B17, B18, B19) or mentions in the notes (i.e., history of chronic hepatitis, established care for chronic hepatitis C). This study was approved by the MedStar Health Research Institute Institutional Review Board.

## 3. Methodology of Risk Prediction Model

### 3.1. Variables

We extracted data from 58 variables (including ICD10 codes) as potential predictors of patients with HCV infection (detectable RNA or reported HCV infection) from structured, semi-structured, and free-text EHR data elements. These variables were identified based on published literature and clinical expertise [6,7]; 46 variables were structured diagnosis codes, one variable was a structured medication code, four were demographic variables, and seven variables were extracted from the semi-structured and free-text data elements (Table 1). The variables determined by clinical experts [DF, AV, and BH] and chart reviews as likely to be captured in clinical notes were identified for NLP modeling. These semi-structured and free-text data elements were patients with a history of substance abuse or use, incarceration, piercings, tattoos, transfusions, needlestick injuries, or sexual transmitted infections [6,7]. The presence of a variable, excluding demographic variables, was coded as 1 and the absence of a variable as 0. Student's *t*-test and the Chi-square test for independence were used to evaluate the significance of continuous and categorical variables, respectively.

**Table 1.** Summary of evaluated risk prediction model ICD10 codes and other variables descriptions with examples.

| Variable (*n*) | ICD10 Codes and Other Variable Descriptions with Examples |
|---|---|
| **Structured Data Elements (47)** | |
| Sexually transmitted infections (14) | A53.9, A54.6, A55, A56.3, A57, A58, A60.1, B00, Z11.4, Z11.3, Z20.6, Z11.59, Z20.2, Z72.5 (Diagnosis) |
| Viral hepatitis and HIV (7) | B17.1, B18.1, B18.2, B19.1, B19.2, Z20.5, B20 (Diagnosis) |
| Substance use disorders (6) | F11, F14, F19.2, K62.89, Z79.899, Z71.5 (Diagnosis) |
| MEDS_opioid | Opioid medication (Medication) |
| Liver disease and laboratory abnormalities (12) | K70, K71, K72, K73, K74, K75, K76, K77, R74.01, R94.5, W46.0, Z01.812 (Diagnosis) |
| Other medical co-morbidities (3) | N18.6, Z99.2, Z94 (Diagnosis) |
| Other factors influencing health status (4) | Z65.1, Z72.89, Z77.21, Z86.59 (Diagnosis) |
| **Semi-structured and free-text data elements (7)** | |
| TEXT_Substance | Patient with history of substance abuse or use including methamphetamine, cocaine, heroin *look up slang* and excludes tobacco, nicotine, and marijuana *Included example "female with history of COPD, depression, substance abuse" Excluded example "Substance Use: Denies"* |

**Table 1.** *Cont.*

| Variable (*n*) | ICD10 Codes and Other Variable Descriptions with Examples |
|---|---|
| TEXT_Incarceration | Patient have been incarcerated<br>*Included example: "Incarcerated 2003 for joyriding"*<br>*Excluded example: "father being incarcerated"* |
| TEXT_Piercing | Presence of piercing on patient<br>*Included example: "Piercing or Tattoo: Professional Piercing'*<br>*Excluded example: "Piercing or Tattoo: No'* |
| TEXT_Tattoo | Presence of tattoo on patient<br>*Included example: "Piercing or Tattoo: Professional Tattoo'*<br>*Excluded example: "Piercing or Tattoo: No'* |
| TEXT_Transfusion | Patient with history of transfusions and solid organ transplants prior to 1992<br>*Included example: "last transfusion on"*<br>*Excluded example: "did not require transfusion"* |
| TEXT_Needlestick | Patient with history of needlestick injury<br>*Included example: "sustained a needlestick to right toe"* |
| TEXT_STI | Patient with history of sexual transmitted infection through homosexual or bisexual behaviors including syphilis, chlamydia, and gonorrhea<br>*Included example "high risk homosexual behavior"* |
| **Demographics (included) (4)**: Sex, age, race, ethnicity | |

### 3.2. Structured Data Elements

Structured data elements are stored in the EHR as discrete elements that map to a unique set of codes or specific quantity values. For this analysis, we used diagnosis codes and medication lists as structured elements. Race, sex, age, and ethnicity are also considered structured data elements.

### 3.3. Semi-Structured Data Elements

Semi-structured data elements are inputted by the user in a checkbox or drop-down, but it gets captured in the EHR as a string. At times, these are fields that are not as defined as structured elements and may also let the user input additional information. These data are converted to a string and are inconsistently captured over time. As a result, text mining approaches are necessary to extract concepts in semi-structured data elements. Semi-structured data elements used in this analysis include extracting concepts from personal histories, for example, drug use, tobacco use, and social history.

### 3.4. Free-Text Data Elements

Free-text data elements correspond to clinician and other provider narratives and notes. This includes narratives associated with a patient's history, a provider's note assessment, and reasons for the visit that are not ICD10-coded. Natural language processing (NLP) algorithms are used to extract concepts from these elements accounting for common spelling mistake (i.e., heroine, opiod).

### 4. Model Development and Validation

We developed and tested models to predict patients with HCV infection. The model was developed and validated following the guidelines for Transparent Reporting of a multivariable Prediction model of Individual Prognosis or Diagnosis (TRIPOD) [16]. A similar approach was previously used to develop and validate an EHR-based HIV-risk prediction tool to identify potential PrEP candidates from which we leveraged relevant ICD10 codes for high-risk sexual behaviors and substance use disorders as well as modeling approach and reporting metrics in our study [17]. We evaluated models with three types of variables: models with structured variables (structured-based model), models with semi-structured and free-text notes variables (text-based model), and models that includes all

variables (full-set model) (Table 1). Each model included four demographic variables (age, sex, race, and ethnicity). As a result, the structured-based models included 51 variables, the text-based models had 11 variables, and the full-set models had 58 variables. We applied each model to three stratifications of data: patients with no history of HCV prior to 2020, patients with a history of treated HCV prior to 2020, and all patients. In total, we had nine test conditions.

We used XGBoost, a decision tree-based boosting ensemble machine learning algorithm, to predict patients with newly identified HCV infection [18]. In a boosting algorithm, many weak learners are trained to correctly predict the observations incorrectly classified in previous training rounds. XGBoost uses a shallow tree as a weak learner and has a good performance in the case of class-imbalanced data classification [19,20]. We evaluated the models using the C-statistic, which is the area under the receiver operating characteristic curve and represents the probability that a randomly drawn HCV-infected patient is ranked as higher risk by the model. Patients identified with HCV infections could either have had a history of treated HCV prior to 2020 or no history of HCV prior to 2020 as these are two distinct groups of patient data. We consider these groupings in the three stratifications of modeling: patients with no history of HCV prior to 2020, patients with a history of treated HCV prior to 2020, and all patients. We used a ten-fold cross-validation to evaluate the generalizability of the models and minimize overfitting. We present the mean and the 95% confident interval (CI) of the C-statistic results from the cross-validation and use the Student's *t*-test to compare performance. Lastly, we calculated the importance of a variable using the average information gain of the variable across all model decision trees. Information gain is the difference in information entropy after a variable split. A higher gain implies more importance. This metric provides a way to look at the additional value that a variable adds to the overall model [21,22]. All analysis, including text extraction, XGBoost modeling, and statistical calculations were carried out with Python 3.9.

## 5. Results

There were 3889 unique patients in the data set. Due to missing demographics information, 325 patients (8.4%) were excluded. The remaining 3564 unique patients were used in the analysis. There were 487 patients (13.7%) who had a newly identified HCV infection and 3077 patients (86.3%) who did not have HCV infection (Table 2). Male patients tended to make up a larger proportion of the newly identified HCV infections compared to female patients (*p* < 0.001). In addition, Black patients made up a larger proportion of newly identified HCV infections compared White and other race patients (*p* < 0.001). The mean age of patients with newly identified HCV infections was older than patients who did not have HCV infection (*p* < 0.001). Lastly, the majority of patients (90.2%) did not have any history of HCV infection prior to 2020.

**Table 2.** Demographic and characteristics of patients by newly identified HCV infection.

| | | Newly Identified HCV Infection Status. No. (%) | | |
|---|---|---|---|---|
| **Characteristic** | **All (N = 3564)** | **Newly Identified HCV Infection (*n* = 487)** | **No Newly Identified HCV Infection (*n* = 3077)** | ***p*-Value** |
| **Sex** | | | | <0.001 |
| Female | 2436 (68.4) | 238 (48.9) | 2198 (71.4) | |
| Male | 1128 (31.6) | 249 (51.1) | 879 (28.6) | |
| **Race** | | | | <0.001 |
| Black | 1617 (45.4) | 256 (52.6) | 1361 (44.2) | |
| White | 1453 (40.8) | 203 (41.7) | 1250 (40.6) | |
| Other | 494 (13.8) | 28 (5.7) | 466 (15.2) | |

**Table 2.** *Cont.*

| Characteristic | Newly Identified HCV Infection Status. No. (%) | | | |
| | All (N = 3564) | Newly Identified HCV Infection (*n* = 487) | No Newly Identified HCV Infection (*n* = 3077) | *p*-Value |
|---|---|---|---|---|
| **Ethnicity** | | | | 0.002 |
| Hispanic | 86 (2.4) | 10 (2.1) | 76 (2.5) | |
| Non-Hispanic | 3241 (90.9) | 462 (94.9) | 2779 (90.3) | |
| Other | 237 (6.6) | 15 (3.1) | 222 (7.2) | |
| **Age** | 45.3 (16.6 SD) | 52.1 (14.5 SD) | 44.3 (16.6 SD) | <0.001 |
| **History of HCV prior to 2020** | | | | <0.001 |
| Yes | 349 (9.8) | 243 (49.9) | 107 (3.4) | |
| No | 3215 (90.2) | 244 (50.1) | 2970 (96.6) | |

## 6. Model Results

The average C-statistics on the structured-based, text-based, and full-set models for all the patients were 0.777 (95% CI: 0.744–0.810), 0.677 (95% CI: 0.631–0.723), and 0.774 (95% CI: 0.735–0.813), respectively. C-statistics are summarized by patient stratification and model variable types in Table 3 and Figure 1. There were no statistically significant differences between the structured-based models and the full-set models. For patients with no history of HCV prior to 2020, there were no significant differences between the three model variable types. For patients with a history of HCV prior to 2020, the text-based model performed worse than both the structured-based and the full-set models; there was no statistically significant difference between the structured-based and full-set models. Text-based models were statistically significantly (*p*-value < 0.001) better for patients with no history compared to patients with a history of HCV prior to 2020.

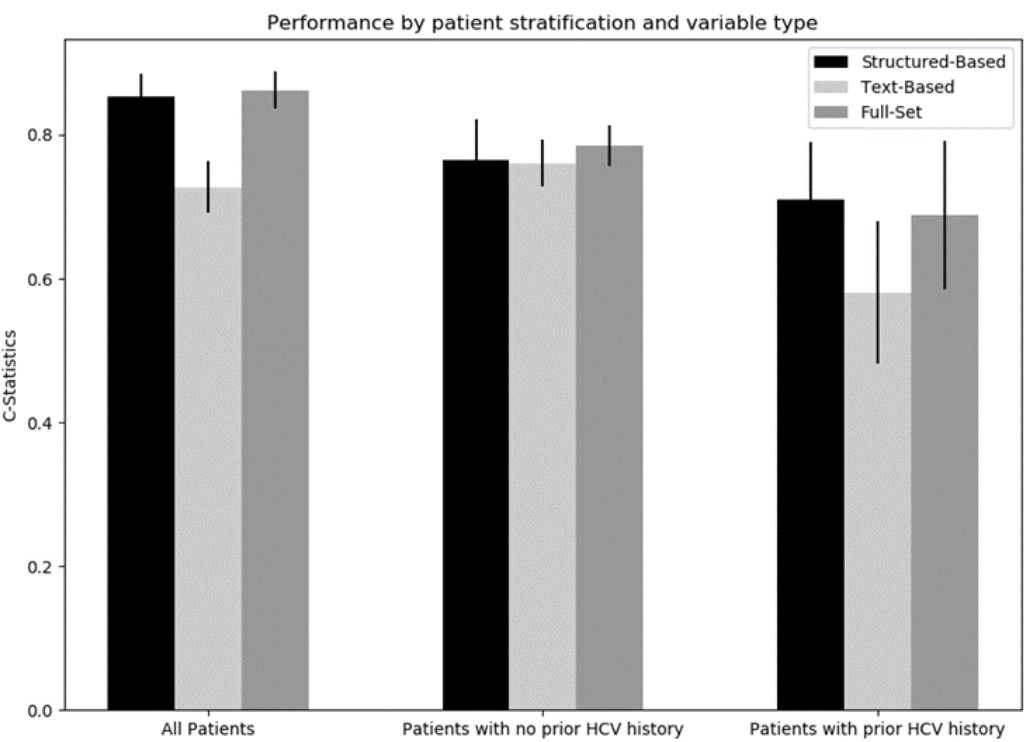

**Figure 1.** Mean of C-statistic performance by patient stratification and model variable type with standard error bars shown.

**Table 3.** Average C-statistics (95% CI) for model and patient stratifications.

|  | **Structured-Based** | **Text-Based** | **Full-Set** |
|---|---|---|---|
| All patients | 0.856 (0.837, 0.876) | 0.725 (0.702, 0.749) | 0.864 (0.845, 0.883) |
| No prior history of HCV | 0.774 (0.735, 0.813) | 0.759 (0.736, 0.781) | 0.780 (0.743, 0.817) |
| Prior history of HCV | 0.700 (0.657, 0.743) | 0.548 (0.478, 0.618) | 0.677 (0.622, 0.732) |
| Average | 0.777 (0.744, 0.810) | 0.677 (0.631, 0.723) | 0.774 (0.735, 0.813) |

## 7. Prevalence of Substance and Opioid Use Amongst Predictors

The most important predictors, as defined by the average gain, in patients *with no* prior history of HCV infection (Table 4; *n* = 3215, middle columns) were opioid related disorders, end stage renal disease, psychoactive substance dependence, and cocaine related disorders. The most important predictors in patients *with* a previously noted history of HCV (Table 4, *n* = 349) were transplanted organ and tissue status, transfusions, liver disease, and end stage renal disease. The full list of predictors is summarized in the Table S1. Oral opioid medication use was prevalent in patients with newly identified HCV infections. The ratio of newly identified HCV infection to no infections amongst patients on opioid medications (0.35) was double the ratio across all patients (0.16). While a ratio provides a general distribution of the data, it will not always correlate to XGBoost model importance. Prior diagnosis codes associated with renal and liver disease or conditions were more important predictors for patients with a prior history of HCV infection than in patients with no HCV infection history.

**Table 4.** Prevalence of important HCV infection predictors ranked by the information gain in the models for patients with no prior HCV infection history.

| Predictor | Predictor Description | All (*n* = 3564) | | | | | No HCV Infection History (*n* = 3215) | | | | | HCV Infection History (*n* = 349) | | | |
|---|---|---|---|---|---|---|---|---|---|---|---|---|---|---|---|
| | | Total | New HCV Infection (*n* = 487) | No new HCV Infection (*n* = 3077) | Importance | Ratio (0.16) | New HCV Infection (*n* = 244) | No new HCV Infection (*n* = 2970) | Importance | Ratio (0.08) | New HCV Infection (*n* = 243) | No New HCV Infection (*n* = 107) | Importance | Ratio (2.27) |
| F11 | Opioid related disorders | 66 | 52 | 14 | 12.471 | 3.71 | 15 | 11 | 11.809 | 1.36 | 37 | 3 | 0 | 12.33 |
| N18.6 | End stage renal disease | 105 | 33 | 72 | 5.318 | 0.46 | 24 | 49 | 6.233 | 0.49 | 9 | 23 | 1.649 | 0.39 |
| Text_Substance | Patient with a history of substance use or abuse | 327 | 146 | 181 | 2.854 | 0.81 | 44 | 146 | 4.492 | 0.30 | 102 | 35 | 0.561 | 2.91 |
| Text_Piercing | Presence of a piercing on patient | 103 | 13 | 90 | 0.492 | 0.14 | 6 | 87 | 2.789 | 0.07 | 7 | 3 | 0.052 | 2.33 |
| F14 | Cocaine related disorders | 16 | 11 | 5 | 2.645 | 2.20 | 2 | 3 | 1.975 | 0.67 | 9 | 2 | 0.59 | 4.50 |
| Z11.3 | Encounter for screening for infections with a predominantly sexual mode of transmission | 454 | 29 | 425 | 1.25 | 0.07 | 7 | 420 | 1.647 | 0.02 | 22 | 5 | 1.088 | 4.40 |
| Meds_Opioid | Opioid medication | 467 | 120 | 347 | 2.295 | 0.35 | 42 | 323 | 1.455 | 0.13 | 78 | 24 | 0.13 | 3.25 |
| Text_STI | Patient with a history of sexual transmitted infection | 1150 | 111 | 1039 | 1.448 | 0.11 | 37 | 1008 | 0.93 | 0.04 | 74 | 31 | 0.411 | 2.39 |
| R94.5 | Abnormal results of liver function studies | 55 | 13 | 42 | 0 | 0.31 | 4 | 36 | 0.721 | 0.11 | 9 | 6 | 0 | 1.50 |
| Z94 | Transplanted organ and tissue status | 84 | 17 | 67 | 15.285 | 0.25 | 8 | 31 | 0.632 | 0.26 | 9 | 36 | 26.074 | 0.25 |
| Z11.59 | Encounter for screening for other viral diseases | 190 | 28 | 162 | 2.029 | 0.17 | 9 | 160 | 0.5 | 0.06 | 19 | 2 | 0 | 9.50 |
| K74 | Fibrosis and cirrhosis of liver | 72 | 34 | 38 | 1.416 | 0.89 | 4 | 16 | 0.474 | 0.25 | 30 | 22 | 2.348 | 1.36 |
| K76 | Other diseases of liver | 93 | 15 | 78 | 2.805 | 0.19 | 0 | 64 | 0 | 0.00 | 15 | 14 | 3.514 | 1.07 |
| Text_transfusion | Patient with a history of transfusions and solid organ transplants prior to 1992 | 622 | 132 | 490 | 1.015 | 0.27 | 39 | 433 | 0 | 0.09 | 93 | 57 | 2.26 | 1.63 |
| Text_tattoo | Presence of a tattoo on patient | 92 | 14 | 78 | 0.836 | 0.18 | 5 | 74 | 0 | 0.07 | 9 | 4 | 0.863 | 2.25 |

## 8. Comparison of Structured and Unstructured Extraction

Text mining was able to identify six risk factors inconsistently coded in structured elements (Table 5): 107 patients had a history of incarceration identified in the free-text, none of whom had the associated ICD10 Z65.1; 11 patients were identified as having a needlestick injury using text mining, only one of which had an ICD10 identifier, W46. Piercings and tattoos were all identified using text mining from the semi-structured and free-text data elements; 352 patients were identified as having substance abuse, 276 from only free-text, 25 from only ICD10 and 51 from both; 1422 patients were identified with sexually transmitted infections, 967 from only free-text, 272 from only ICD10, and 183 from both.

**Table 5.** Prevalence and percent distribution of identified six risk factors in structured and unstructured data elements.

| Text Predictor (Associated ICD10) | Total Patients Identified | Identified Only from Free-Text (%) | Identified Only from ICD10 (%) | Identified from Both (%) |
|---|---|---|---|---|
| TEXT_Incarceration (Z65.1) | 107 | 107 (100) | 0 | 0 |
| TEXT_Needlestick (W46) | 11 | 10 (91) | 0 | 1 (9) |
| TEXT_Substance (Z71.5, F11, F14, F19.2) | 352 | 276 (78) | 25 (7) | 51 (14) |
| TEXT_STI (A53.9, A54.6, A55, A56.3, A57, A58, A60.1, B00, B17.1, B18.1, B18.2, B19.1, B19.2, B20, K62.89) | 1422 | 967 (68) | 272 (19) | 183 (13) |
| TEXT_Piercing (Z41.3) | 103 | 103 (100) | 0 | 0 |
| TEXT_Tattoo (L81.8) | 92 | 92 (100) | 0 | 0 |

## 9. Discussion

### 9.1. Utility of Structured Fields

The full-set and structured-based models had comparable performance statistics and the best overall model performance. We suspect that the structured-based model had improved performance over text-based models alone because the scope of data captured by the various ICD10 codes are enough to have high performance results in this data at a large healthcare system. It is important to note that certain structured fields were included in the model, such as opioid medications and sexually transmitted infections, which are not listed in the CDC and AASLD/IDSA risk factor recommendations. Having a model that relies only on structured fields is very useful as such models can be more easily shared between, and implemented by, different healthcare practices and systems. By utilizing and leveraging platforms such as the Observational Medical Outcomes Partnership (OMOP) Common Data Model (CDM), [23] models based on only structured data elements can be more easily shared and tested for generalizability.

### 9.2. Free-Text Concepts

Using NLP features extracted from clinical documentation in a risk-prediction model offers a novel means to identify persons at high risk for newly identified HCV with the need for repeat HCV testing, and for targeted navigation and behavioral health interventions. There was no statistically significant difference between models for patients with no HCV infection history, demonstrating the utility of NLP for patients with no prior history of HCV. It is important to note the differences between text extraction and the ICD10 codes as identified in Table 5, specifically for sexually transmitted infections and substance use/abuse, as this demonstrates that text mining captures a wider net of patients which could explain the comparable performance, even given a limited predictive variable set. Consistently mapping structured fields across resource limited healthcare systems can be difficult, especially without resources such as OMOP CDM. However, having a text-based model with similar performances for patients with no HCV infection history, provides an

alternative opportunity to perform limited HCV infection prediction without the reliance of provider-driven output of structured fields.

*9.3. HCV and Opioid Use*

Opioid prescribed medication and use predictors were consistently among the leading predictors in all patient groups and strongest in patients with no HCV infection history. Results from this study support many other studies that considers injection opioid use as a concern for HCV transmission and thus incorporated in the CDC and AASLD/IDSA risk factors [2,6,24,25]. Although rates of prescribing opioids have decreased in recent years in attempts to curb the opioid epidemic, prescription opioids remain a significant risk factor for the development of opioid use disorder and injection drug use (IDU); the latter is the single most significant risk factor for acquiring HCV [25,26]. Despite the established trajectory from oral prescription opioids to opioid use disorder, then to IDU, and IDU to HCV, oral prescribed opioids are not yet included in defined CDC and AASLD/IDSA risk factors. Previous work from our group associated this as a factor in HCV infection [27]. Our results add to the literature and make an additional case for including oral prescribed opioids in the risk factor guidelines.

*9.4. Patient Care*

While universal testing is recommended, it is possible that patients might still 'fall through the cracks' or be missed as it is not certain that such recommendations will be required or will be followed up. Integrating this model that predicts patients at-risk for HCV infection into the clinical workflow may help providers prioritize patients who are at high-risk and who they should follow more closely. It may also make the compelling argument for systems to allocate increased resources for linkage into care. Identifying patients at higher risk is helpful for providers and healthcare facilities when prioritizing resources especially in clinics that are more resource constrained. In our continuing work, the risk-prediction model will be integrated into a clinical decision support (CDS) tool for clinicians. This will test, validate, and operationalize the model, as well as provide mechanisms for the model to learn from user feedback.

## 10. Limitations

Data used in this study come from a single healthcare system and the variables sections were based on chart reviews from the healthcare system. Although the healthcare system consists of a diverse collection of hospitals and ambulatory sites, evaluating the generalizability of these models at different healthcare and other systems will be important. Our model did not include pregnancy as a specific risk factor variable; however, it is recommended that women have an HCV test with each pregnancy and this will be built into the CDS tool. These results rely on the use of large data and artificial intelligence which does not allow for granular review. However, the use should never supersede the need for continued and sound clinical judgement. Lastly, understanding ethnical impacts and consideration for ML and NLP models is critical for building reliable and equitable tools and techniques for all patients and care providers. This includes further investigating all dimensions of fairness, such as accurately obtaining patient race and ethnicity information, as to not propagate potential systemic biases in how data were collected or modeled.

## 11. Conclusions

Integrating a structured-based or full-set risk-prediction model into the clinical workflow may assist providers prioritize patients, and help systems provide resources for patients who are at high risk for HCV infection and re-infection. We demonstrated the comparable performance of text-based and structured-based models for patients with no HCV infection history. In addition, the models highlight the importance of oral prescribed opioids for predicting HCV infection suggesting the utility of including oral prescribed opioids as a risk factor in guidelines. Using NLP features extracted from clinical documen-

tation in a risk-prediction model, when available, offers a novel means to capture predictors often missed in structured data elements.

**Supplementary Materials:** The following supporting information can be downloaded at: https://www.mdpi.com/article/10.3390/gidisord5020012/s1. Table S1. The prevalence, ratios, importance (Imp.) and descriptions of all predictors amongst the different patient subsets. (Note: Importance of '–' indicates the variable was not used the model splits).

**Author Contributions:** Conceptualization, A.F., B.H., M.B., S.S.H. and D.F.; methodology, A.F., J.H., S.G., B.H., M.B., S.S.H., A.V., S.F. and D.F.; software, A.F., J.H. and S.F.; validation, A.F., S.G., B.H., M.B., S.S.H., A.V. and D.F.; formal analysis, A.F., J.H., S.G., B.H., M.B., S.S.H., A.V., S.F. and D.F.; investigation, A.F., J.H., S.G., B.H., M.B., S.S.H., A.V., S.F. and D.F.; resources, A.F., J.H., S.G., B.H., S.F. and D.F.; data curation, A.F., J.H., S.G., B.H., M.B., S.F. and D.F.; writing—original draft preparation, A.F., J.H., S.G., B.H., M.B., S.S.H., A.V., S.F. and D.F.; writing—review and editing, A.F., J.H., S.G., B.H., M.B., S.S.H., A.V., S.F. and D.F.; visualization, A.F.; supervision, A.V. and D.F.; funding acquisition, D.F. All authors have read and agreed to the published version of the manuscript.

**Funding:** Investigator-initiated study grant funding from Gilead Sciences, Inc. (Contract # IN-US-987-6133). This work was presented in part as a poster presentation at The Liver Meeting, Washington DC, 2022.

**Institutional Review Board Statement:** This study was approved by the MedStar Health Research Institute's Institutional Review Board: approval number 3865.

**Informed Consent Statement:** Not applicable.

**Data Availability Statement:** Data analyzed in this study is available here: https://journals.plos.org/plosone/article?id=10.1371/journal.pone.0279972#sec017 (accessed on 5 May 2022).

**Conflicts of Interest:** The authors declare no conflict of interest.

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
