# Peer review of "Evaluation of Structured, Semi-Structured, and Free-Text Electronic Health Record Data to Classify Hepatitis C Virus (HCV) Infection"

_gastrointestdisord, doi:10.3390/gidisord5020012_

Round 1

Reviewer 1 Report

In 2020, the CDC updated their guidance on HCV screening, especially for testing those at high risk of infection. The proper identification and diagnosis of HCV patients is critical to achieve the goal of global HCV elimination by 2030 and can be improved by the training of a model with both CDC and non-CDC risk factors as variables. In this manuscript, Fong et al. generate several models for HCV risk prediction by using machine learning on electronic health record (EHR) data and risk factors identified through text mining. The authors found that all models showed similar predictive performance, though the most important risk factors appeared to depend on the HCV infection history of training data. Overall, these models can provide a useful framework for identifying patients with a high HCV infection risk.

Strengths:

1.       The paper is well-written throughout, and is especially clear in its description of the various data elements and risk factors used. The examples shown in Table 1 are quite helpful for orienting readers.

2.       Though they are somewhat redundant by nature, Figure 1 and Table 3 show the key results of this paper in a clear and concise manner.

To improve the paper, more detail about the study design rationale, choice of variables, and validity of generated models is warranted. Specific comments/questions are below, separated by section.

Introduction:

1.       Page 2 Line 46: “who at high risk” typo

Methods:

1.       Page 3 Lines 10-11: what percentage of potential data did you have to exclude because of missing demographic information about patients? Could this data exclusion have skewed models in some way?

2.       Page 4 Lines 17-20: Do you have to take possible typos into account for free-text data elements? If so, do you only include exact matches or common typos as well?

3.       Page 4 Lines 25-26: How did the HIV EHR data models help detection, and do you expect a similar effect with a HCV model?

4.       Page 5 Lines 13-15: This sentence mentions the “probability that a randomly drawn HCV-infected patient was ranked higher” – this specifically refers to predicting a newly infected HCV patient, and not a patient with a previous history of HCV infection? Would be helpful to clarify and emphasize that new infection and previous history are two overlapping but distinct groups of patient data.

5.       Page 5. Line 22: “Analysis was done in Python 3.9” – does this only refer to C-statistic calculations? If so, what program was used to generate models with XGBoost?

Table 1:

1.       Why were some specific ICD10 codes included (e.g. A54.6) while other codes that are very similar were excluded? What criteria were used to include ICD10 codes? More detail about the rationale of including specific variables would be helpful to add to the methods.

Table 2:

1.       Is the distribution of cases in the training data representative of new HCV cases overall (e.g. 50/50 male to female, only 2% hispanic, etc.)?

2.       What method was used to calculate the listed p-values?

Results:

1.       Page 11 Lines 1-10: If free-text mining added data, why did the structured-only model show improved predictive performance over text-based alone? Was the model better without that additional data, or was the data combined during training? Somewhat addressed in the discussion, but more detail could be helpful.

Table 4:

1.       On Page 7 lines 10-11, the authors mention a higher ratio of new HCV infections for patients with an opioid-related variable, which is shown in Table 4 with a relatively high importance. Does a higher ratio in this table suggest that a particular variable is more likely to be an important predictor? If so, why was the F11 predictor not important for patients with a history of HCV infection even though the ratio is quite high? The ratio information is interesting, but additional detail on why these ratios are important would be useful to include.

Discussion:

1.       Page 11 lines 32-33: “models and full-set and structure-based” – the wording of this sentence makes it difficult to follow the significant difference (or differences) that you are highlighting. This section would be helped by making this sentence clearer.

2.       Related to Figure 1/a few tables – why would predictive performance for patients with previous HCV history be lower than for patients with no infection history? Is the difference largely a function of the 10-fold difference in patient data available?

3.       Page 12 lines 7-9: What would need to be done to make this statement true for a predictive model for patients with HCV infection history? Does this statement only apply to patients with no infection history?

4.       Page 12 lines 20-22: What’s the predictive performance of the model if opioid variables were removed? This could make the case for including opioids even clearer, though perhaps highlighting the importance of opioid-related variables would be enough.

5.       Page 12 lines 37-44: Do you plan to assess the accuracy of these models using patient data independent of what was used for model generation? Though cross-validation was performed, is the real-world generalizability limited by not challenging these models on an independent test set?

Supplemental:

1.       A minor point, but not all of the ratios in the table have the same formatting (e.g. some are percentages instead), and it would help to make it consistent across the table.

Author Response

  1. Page 2 Line 46: “who at high risk” typo

>>Sentence corrected to state ‘…patients at high risk…’

Methods:

  1. Page 3 Lines 10-11: what percentage of potential data did you have to exclude because of missing demographic information about patients? Could this data exclusion have skewed models in some way?

>>Approximately 8.4% of the original unique patients (325 out of 3889) had to be excluded because of missing demographics information. This could potentially skew the model as it could reflect systemic data collection challenges. The results was updated to reflect the amount of missing patients. “There were 3889 unique patients in the data set. 325 patients (8.4%) were excluded to due missing demographics information. The remaining 3564 unique patients were used in the analysis.”

In addition, the discussion was expanded to describe the challenges with how data is collected  “Lastly, understanding ethnical impacts and consideration for ML and NLP models is critical for building reliable and equitable tools and techniques for all patients and care providers. This includes further investigating all dimensions of fairness, such as accurately obtaining patient race and ethnicity information, as to not propagate potential systemic biases in how data were collected or modeled.”

  1. Page 4 Lines 17-20: Do you have to take possible typos into account for free-text data elements? If so, do you only include exact matches or common typos as well?

>>We included common typos as our analysis. We updated the methods section to include “…extract concepts from these elements accounting for common misspellings (ie heroine, opiod) .”

  1. Page 4 Lines 25-26: How did the HIV EHR data models help detection, and do you expect a similar effect with a HCV model?

>>We leveraged the approach Marcus et al. took in developing the HIV EHR model by using similar variables such as ICD10 codes for high-risk sexual behaviors and substance use disorders, modeling methods (10-fold cross-validation) and reporting metric AUC C-statistic. We also contacted and had a helpful discussion with Dr. Marcus to review her approach and her thoughts on it’s expansion for HCV modeling.

The methods: model development and validation section was updated to clarify the following: “A similar approach was previous used to develop and validate an EHR-based HIV risk prediction tool to identify potential PrEP candidates from which we leveraged relevant ICD10 codes for high-risk sexual behaviors and substance use disorders as well as modeling approach and reporting metrics in our study.(17)”

  1. Page 5 Lines 13-15: This sentence mentions the “probability that a randomly drawn HCV-infected patient was ranked higher” – this specifically refers to predicting a newly infected HCV patient, and not a patient with a previous history of HCV infection? Would be helpful to clarify and emphasize that new infection and previous history are two overlapping but distinct groups of patient data.

>>Models were developed to predict newly infected HCV in patients with no history of HCV infection. Models were also built to predict HCV reinfections in patients with a history of treated HCV. We expanded the methods text to clarify the distinct patient data and model development process. “Patients identified with HCV infections could either have had a history of treated HCV prior to 2020 or no history of HCV prior to 2020 as these are two distinct groups of patient data. We consider these groupings in the three stratifications of modeling: patients with no history of HCV prior to 2020, patients with a history of treated HCV prior to 2020, and all patients.”

  1. Page 5. Line 22: “Analysis was done in Python 3.9” – does this only refer to C-statistic calculations? If so, what program was used to generate models with XGBoost?

>>All modeling and statistical analysis was done in Python 3.9.

The method text was updated to “All analysis, including text extraction, XGBoost modeling, and statistical calculations was done using Python 3.9.”

Table 1:

  1. Why were some specific ICD10 codes included (e.g. A54.6) while other codes that are very similar were excluded? What criteria were used to include ICD10 codes? More detail about the rationale of including specific variables would be helpful to add to the methods.

>> The variables determined by clinical experts [DF, AV, BH] and chart reviews and informed by HCV guidelines and prior research. (6, 7) This approach is limited by the charts reviewed. We added this limitation to the discussion section. “Data used in this study comes from a single healthcare system and the variables sections were based on chart reviews from the healthcare system.”

Table 2:

  1. Is the distribution of cases in the training data representative of new HCV cases overall (e.g. 50/50 male to female, only 2% hispanic, etc.)?

>>The reviewer is correct; the distribution of cases in the training data was representative of all the overall distribution as the training data was equally sampled.

  1. What method was used to calculate the listed p-values?

>>Student t-tests and Chi-square tests for independence were used to calculate the p-values for continuous and categorical variables respectively. The methods variable section was updated with the following for clarification “Student t-tests and Chi-square tests for independence were used to evaluate the significance of continuous and categorical variables respectively.”

Results:

  1. Page 11 Lines 1-10: If free-text mining added data, why did the structured-only model show improved predictive performance over text-based alone? Was the model better without that additional data, or was the data combined during training? Somewhat addressed in the discussion, but more detail could be helpful.

>> When all the data was combined in training, the model had the best performance. However, this performance was only marginally better then the structured-only model. We believe the structured-only model generally had better predictive performance over text-based alone because the scope of data captured by the various ICD10 fields is enough to have high performance results in this data at a large healthcare system. The discussion was updated to expand with the following “We suspect that the structured-based model had improved performance over text-based models alone because the scope of data captured by the various ICD10 codes are enough to have high performance results in this data at a large healthcare system.”

Table 4:

  1. On Page 7 lines 10-11, the authors mention a higher ratio of new HCV infections for patients with an opioid-related variable, which is shown in Table 4 with a relatively high importance. Does a higher ratio in this table suggest that a particular variable is more likely to be an important predictor? If so, why was the F11 predictor not important for patients with a history of HCV infection even though the ratio is quite high? The ratio information is interesting, but additional detail on why these ratios are important would be useful to include.

>>The ratio was included to give the readers an sense of what was in the data as the XGBoost importance, as defined as average gain, and ratio will not necessarily correlate as XGBoost is a non-deterministic modeling approach. The results on Page 7 is updated with the clarification “While ratio provides a general distribution of the data, it will not always correlate to XGBoost model importance.”

Discussion:

  1. Page 11 lines 32-33: “models and full-set and structure-based” – the wording of this sentence makes it difficult to follow the significant difference (or differences) that you are highlighting. This section would be helped by making this sentence clearer.

>>Sentence was changed to “There was no statistically significant difference between models for patients with no HCV infection history”

  1. Related to Figure 1/a few tables – why would predictive performance for patients with previous HCV history be lower than for patients with no infection history? Is the difference largely a function of the 10-fold difference in patient data available?

>>We believe the difference in performance is likely a result of how the patient data was stratified which could also be impacted by the cross-fold validation as it will result in fewer positive patients to train from.

  1. Page 12 lines 7-9: What would need to be done to make this statement true for a predictive model for patients with HCV infection history? Does this statement only apply to patients with no infection history?

>>This statement could apply to both patients with no infection history and patients with infection history. However, the results show that the most benefit could be for patients with no infection history. The discussion was updated with this clarification: “having a text-based model with similar performances for patients with no HCV infection history, provides an alternative opportunity to perform limited HCV infection prediction without the reliance of provider-driven output of structured fields.”

  1. Page 12 lines 20-22: What’s the predictive performance of the model if opioid variables were removed? This could make the case for including opioids even clearer, though perhaps highlighting the importance of opioid-related variables would be enough.

>>We agree with the reviewer that comparing models without opioids would be interesting for further work and would make the case for including opioids even clearer. We believe that the current modeling approach captures the importance of opioids as XGBoost randomly selects variables to include in the model and prunes those that are not useful. It would be interesting to future testing of models with and without opioids to understand the predictive impact of opioids.

  1. Page 12 lines 37-44: Do you plan to assess the accuracy of these models using patient data independent of what was used for model generation? Though cross-validation was performed, is the real-world generalizability limited by not challenging these models on an independent test set?

>>The reviewer highlights a very important consideration on an independent test set in real-world testing which could also include missing, incomplete, or incorrect data. We are currently working on addresses these limitations as we work to implement these models in-situ.

Supplemental:

  1. A minor point, but not all of the ratios in the table have the same formatting (e.g. some are percentages instead), and it would help to make it consistent across the table.

>>Table updated to the same format to be consistent as ratios.

Reviewer 2 Report

This is a very interesting and innovative manuscript that uses natural language processing in screening for HCV. The methodology is well articulated in this study and the authors have done a thorough interrogation of the probability of diagnosis of HCV based on semi structured and free text elements.

Some minor points:

1. In Table 1 the diagnoses are listed as codes. Are these ICD-10 codes>? if so that should be expressly stated in the table in addition to the text.

2. The authors should elaborate with references on how they chose the free text data elements. What data elements were excluded based on published HCV etiologies?

3. The development of predictors for HCV infection based on NLP does come with significant ethical issues. The authors might want to elaborate on this in the discussion.

Author Response

  1. In Table 1 the diagnoses are listed as codes. Are these ICD-10 codes>? if so that should be expressly stated in the table in addition to the text.

>>The codes listed are ICD-10 codes. Table 1 and the corresponding text for Table 1 were updated to indicate ICD-10 codes.

  1. The authors should elaborate with references on how they chose the free text data elements. What data elements were excluded based on published HCV etiologies?

>> The variables determined by clinical experts [DF, AV, BH] and chart reviews as likely to be captured in clinical notes were identified for NLP modeling. These semi-structured and free-text data elements were patients with a history of substance abuse or use, incarceration, piercings, tattoos, transfusions, needlestick injuries, or sexual transmitted infections. Elements that could not be readily identified in chart reviews were excluded from the free-text data elements.

The method variable section was clarified with the following:

“The variables determined by clinical experts [DF, AV, BH] and chart reviews as likely to be captured in clinical notes were identified for NLP modeling. These semi-structured and free-text data elements were patients with a history of substance abuse or use, incarceration, piercings, tattoos, transfusions, needlestick injuries, or sexual transmitted infections.(6, 7)”

  1. The development of predictors for HCV infection based on NLP does come with significant ethical issues. The authors might want to elaborate on this in the discussion.

>>We agree that ethnical issues such as modeling biases needs to be considered to understand both the limitations of NLP and ML models and the underlying data as to not propagate potential systemic biases in how data was collected or modeled. Understanding ethnical impacts and consideration for machine learning and language model is critical for building reliable and equitable tools and techniques for all patients and care providers. The limitations section in the discussion was expanded with the following: “Lastly, understanding ethnical impacts and consideration for ML and NLP models is critical for building reliable and equitable tools and techniques for all patients and care providers. This includes further investigating all dimensions of fairness, such as accurately obtaining patient race and ethnicity information, as to not propagate potential systemic biases in how data were collected or modeled.”

Reviewer 3 Report

Fong et al assessed the performance of three models to predict patients with HCV infection. They found that the overall performance of full-set and structured-based models had better diagnostic accuracies than text-based model.

1.     Abstract: I suggested that the authors used non-structured illustration in the abstract by removing Introduction, Methods, Results and Conclusion section. The conclusion was confusing, since the results did not mentioned any data of the three models for patients with no history of HCV before 2020. The authors should provide illustration for Table 3 in the abstract to support to conclusion made by the authors. Please rewrite the abstract to meet my request.

2.     Introduction: The use of subheading in the introduction was strongly discouraged. Please make paragraphs instead of the use of subheadings in the Introduction (HCV screening, Leveraging the Electronic Health Record (EHR) in HCV Prediction).

Author Response

  1. Abstract: I suggested that the authors used non-structured illustration in the abstract by removing Introduction, Methods, Results and Conclusion section. The conclusion was confusing, since the results did not mentioned any data of the three models for patients with no history of HCV before 2020. The authors should provide illustration for Table 3 in the abstract to support to conclusion made by the authors. Please rewrite the abstract to meet my request.

>>The Introduction, Methods, Results, and Conclusion sections were removed from the abstract and an illustration for Table 3 was included in the abstract to support the conclusion.

The abstract was updated with “The full-set model performed slightly better than the structured-based model and similar to text-based models for patients with no history of HCV prior to 2020, average C-statistics of 0.780, 0.774, and 0.759 respectively.”

  1. Introduction: The use of subheading in the introduction was strongly discouraged. Please make paragraphs instead of the use of subheadings in the Introduction (HCV screening, Leveraging the Electronic Health Record (EHR) in HCV Prediction).

>>Subheadings in the introduction were removed and replaced with paragraphs.